# Uptake of Magnetite Nanoparticles on Polydopamine Films Deposited on Gold Surfaces: A Study by AFM and XPS

**DOI:** 10.3390/nano14211699

**Published:** 2024-10-24

**Authors:** Andrea Atrei, Shalva Chokheli, Maddalena Corsini, Tóth József, Giuseppe Di Florio

**Affiliations:** 1Dipartimento di Biotecnologie, Chimica e Farmacia, Università di Siena, 53100 Siena, Italy; s.chokeli@student.unisi.it (S.C.); maddalena.corsini@unisi.it (M.C.); 2HUN-REN Institute for Nuclear Research, H-4026 Debrecem, Hungary; toth.jozsef@atomki.hu; 3ENEA-Italian National Agency for New Technologies, Energy and Sustainable Economic Development, Casaccia Research Centre, 00124 Rome, Italy; giuseppe.diflorio@enea.it

**Keywords:** magnetite, nanoparticles, coatings, polydopamine, AFM, XPS

## Abstract

Polydopamine has the capacity to adhere to a large variety of materials and this property offers the possibility to bind nanoparticles to solid surfaces. In this work, magnetite nanoparticles were deposited on gold substrates coated with polydopamine films. The aim of this work was to investigate the effects of the composition and morphology of the PDA layers on the sticking of magnetite nanoparticles. The polydopamine coating of gold surfaces was achieved by the oxidation of alkaline solutions of dopamine with various reaction times. The length of the reaction time to form PDA was expected to influence the composition and surface roughness of the PDA films. Magnetite nanoparticles were deposited on these polydopamine films by immersing the samples in aqueous dispersions of nanoparticles. The morphology at the nanometric scale and the composition of the surfaces before and after the deposition of magnetite nanoparticles were investigated by means of AFM and XPS. We found that the amount of magnetite nanoparticles on the surface did not vary monotonically with the reaction time of PDA formation, but it was at the minimum after 20 min of reaction. This behavior may be attributed to changes in the chemical composition of the coating layer with reaction time.

## 1. Introduction

Polydopamine (PDA) has been the object of a large number of investigations motivated by the capacity of PDA to coat surfaces of various types of materials [1,2,3,4,5]. Films of PDA are deposited either by the oxidation of dopamine (DA) or electrochemically in the case of conductive substrates [6,7]. The formation of PDA is triggered by the oxidation of DA, and PDA contains several building blocks (see Figure 1). However, the reaction mechanisms leading to the formation of PDA and its structure are not completely determined yet [8,9]. PDA films and particles with tailored properties are obtained by varying the preparation method and conditions (DA conc., pH, type of buffer, oxidant, etc.) [2]. PDA films can be functionalized with (bio)molecules by exploiting the functional groups present at the surface [10]. Functionalization is particularly relevant for potential applications in environmental and biomedical fields [11,12,13]. PDA films offer a versatile method for modifying the chemical and physical characteristics of solid surfaces by enabling the attachment of nanoparticles (NPs). There are two key approaches to achieve this goal: (a) first coating the NPs with a PDA film and subsequently anchoring them onto a clean surface or (b) forming a PDA layer directly on a substrate before depositing the NPs. Both strategies exploit the adhesive properties of PDA, making it a powerful tool for surface modification and functionalization. Among the various types of NPs, magnetic NPs coated with PDA have received a lot of attention [14,15,16]. Fe_3_O_4_ (magnetite) and γ-Fe_2_O_3_ (maghemite) NPs anchored to solid surfaces by means of PDA layers may have potential applications in electrocatalysis and as electrochemical sensors [17,18]. Studies performed by Atomic Force Microscopy (AFM) showed that PDA on gold and other substrates formed 2D islands with thickness and surface coverage increasing with deposition time [19]. On top of these 2D islands, features with various heights (ranging from a few nm [19] to hundreds of nm [20,21], depending on the reaction conditions) were observed in AFM images. As a result, rather rough surfaces were obtained. It is not clear if these grains were due to PDA nanoparticles formed in the solution that stuck to the growing film or if they were the results of reactions at the solid–solution interface [22,23]. Variation of the reaction time should result in changes in the composition and surface roughness of PDA films, and both are expected to influence the adhesion of NPs on these surfaces. To achieve a better understanding of the interplay of these two effects, in the present work, we investigated by means of AFM and X-ray Photoelectron Spectroscopy (XPS) the uptake of Fe_3_O_4_ NPs on PDA films deposited on gold substrates. PDA/Au films were prepared for increasing reaction times. The aim of this study was to determine how the morphology at the nanometric scale and the composition of the PDA films affected the sticking of Fe_3_O_4_ NPs on them. AFM images allowed us to determine the effect of the reaction time on the morphology of the PDA layer and the variations after immersion in the dispersion of Fe_3_O_4_ NPs. XPS measurements provided the chemical information (which could not be directly obtained by AFM images) needed to monitor both the composition of the PDA layer and the uptake of the Fe_3_O_4_ NPs.

## 2. Materials and Methods

### 2.1. Materials

All materials were research-grade products and used as received. Dopamine hydrochloride (purity 99%) was purchased from Fisher Scientific (Milano, Italy). Pieces (approximately 10 mm × 5 mm), cut from a gold-coated silicon (p-type) wafer oriented along the (100) plane (Electron Microscopy Sciences, Hatfield, PA, USA), were used as substrates to deposit the PDA films.

### 2.2. Preparation of Samples

Fe_3_O_4_ NPs were prepared by coprecipitation from solutions of FeCl_3_⋅6H_2_O and Fe(NH_4_)_2_(SO_4_)_2_⋅6H_2_O (2:1 Fe(III)/Fe(II) molar ratio) in double-distilled water (DDW) by the adding of NaOH. Details about the preparation and characterization of the Fe_3_O_4_ NPs are reported elsewhere [24,25]. In these papers, particle sizes, particle size distribution, and composition are reported. PDA films were deposited on the gold substrates by immersing the gold-coated silicon pieces in a H_2_PO_4_^−^/HPO_4_^2−^ (total phosphate ion concentration 0.05 M) buffer solution at pH 8.0 containing 1 mg/ml of dopamine hydrochloride. The reaction was carried out in a beaker (to guarantee contact with air) under shaking at room temperature (RT), with reaction times of 10, 20, and 60 min and 24 h. After the selected reaction time was reached, the sample was rinsed with DDW and dried under a flux of N_2_. Fe_3_O_4_ NPs were deposited on clean and PDA-covered Au substrates by immersing the gold-coated silicon pieces in a dispersion (1 mg/mL) of magnetite NPs in DDW at a pH close to 6.5. Before the immersion of the gold substrates, the dispersions of NPs were sonicated for ca. 20 min. The gold substrates were kept immersed in the dispersion of NPs for 1 h under shaking at RT. After this treatment, the samples were washed with DDW to remove the NPs that did not adhere to the substrate. The samples were dried under N_2_ flux.

### 2.3. AFM and XPS Measurements

AFM images were collected in semi-contact mode by using a Solver P47-PRO SPM (NT-MDT, Zenelograd, Russia) microscope equipped with a silicon tip. The force constant of the cantilever was ca. 20 N/m and the oscillating frequency was 200 kHz. AFM images were analyzed by using the software of the microscope to determine height profiles, average roughness (Ra), and root mean square roughness (Rq) [26]. For each sample, the reported data were the average of 3–5 images acquired in different zones. AFM images were also analyzed by using the persistent homology (PH) method [27,28]. Being a tool of topological data analysis, PH allows one to study the geometrical structures of an AFM image, avoiding the arbitrariness of several conventional approaches [27]. The TDA package for topological data analysis in the RStudio development environment was employed [29]. Data visualization and the post-processing of data were implemented in the Python JupyterLab development environment with an in-house script.

XPS measurements were performed in an ultra-high vacuum chamber equipped with a homemade hemispherical electron energy analyzer and a non monochromatized Al Kα X-ray source. The angle between the direction of the photon beam and the axis of the analyzer was 70°. Photoelectrons were collected in the direction normal to the surface. The XPS spectra were acquired in the fixed retarding ratio (FRR) mode. The binding energy (BE) scale was calibrated by setting the aliphatic component of the C1s peak to 285.0 eV. For background subtraction (Shirley type) and curve fitting analysis, XPSPEAK 4.1 software was used.

## 3. Results and Discussion

### 3.1. AFM Results

AFM images of PDA films on gold surfaces prepared at reaction times of 10, 20, and 60 min and 24 h (1440 min) are shown in Figure 1a–d, left column. On the right column of Figure 1a–d, AFM images acquired for the same samples after immersion in the dispersions of Fe_3_O_4_ NPs are shown. AFM images of the gold substrate before PDA film deposition exhibited features with a height below 5 nm and roughness values of around 1 nm (Appendix A). Upon increasing the reaction time, for the PDA films, there was a progressive increase in the number of islands and their heights. The average height and Ra and Rq values increased monotonically with the reaction time, reaching limiting values (Figure 2).

These observations suggest that PDA films did not grow on Au, forming islands of a constant thickness and uniformly covering the whole surface, in an ideal layer-by-layer growth mode. The granular features observed by AFM could have been due either to PDA NPs formed in solution that adhered to the growing films or by the reaction at the solid–solution interface. The variations in surface morphology after immersion of the PDA/Au films in the dispersions of Fe_3_O_4_ NPs revealed the uptake of NPs on the polymer film (Figure 1a–d, right column). It is important to note that Fe_3_O_4_ NPs did not adhere to the clean Au substrate, as verified by AFM (see Appendix A). For each reaction time, an increase in the average height, Ra, and Rq of the surface after immersion in the dispersion of Fe_3_O_4_ NPs was found (Figure 2). The trend of average height, Ra, and Rq with respect to reaction time, in the case of PDA/Au films with Fe_3_O_4_ NPs, appeared to be more complex than that of the PDA/Au films. For instance, those parameters were larger for the 10 min sample than for all the other samples (Figure 2). This could be attributed to the presence of few aggregates of NPs with relatively great heights in the case of the 10 min sample, which were almost absent in the other samples. After 60 min of reaction, the PDA/Au film appeared to be more uniformly covered by Fe_3_O_4_ NPs compared to the 10 and 20 min samples (Figure 1c, right column). In the AFM images of the 60 min sample, there were only relatively few islands higher than the average height, and the areas around were filled with smaller aggregates of NPs. A possible explanation of these results is the following. In the first 10 min of the reaction, the formation of PDA occurred to a limited extent, and the gold surface was covered mostly by adsorbed DA. Fe_3_O_4_ NPs were capable of binding to this surface, thus producing the increase in surface roughness compared to PDA/Au surfaces. After 20 min of reaction, relatively few PDA islands formed, with the largest fraction of the gold surface being covered by intermediate species (see Figure 1). We can hypothesize that Fe_3_O_4_ NPs adhere only to PDA islands already formed, whereas they do not stick on the areas covered by reaction intermediates. Fe_3_O_4_ NPs bonded to the PDA islands appeared in the AFM images as the isolates of features with the greatest heights. After 60 min of reaction, the whole gold surface was covered by the PDA film, on which Fe_3_O_4_ NPs were capable of adhering, forming a more uniform layer.

PH analysis helped to discriminate between the contributions to the surface morphology of the PDA film and the deposited Fe_3_O_4_ NPs, as shown in Figure 3. PH is a graphical way to display topological results, known as a persistent diagram (PD). In this diagram, the topological features are displayed as points in a scatter plot. The coordinates of each point correspond to the height in the AFM image at which the surface feature was revealed by the PH filtration process (birth) and to the height of disappearance of that feature (death) [27,28]. Therefore, the difference between “birth” and “death” returns a measure of the surface feature height (persistence of the topological feature). Graphically, this corresponds to the distance of the point from the bisector. For a surface, there are two homology generators, one of 0 dimensionality and one of 1 dimensionality, and, for both, it is possible to build a relative PD. In the 0-dimensional PD are displayed topological features attributable to reliefs of the surfaces (hills, grains, etc.), while, in the 1-dimensional PD, pits enclosed in the surface landscape, like valleys and basins, are represented. Different from many conventional approaches used to characterize surfaces, often relying on single parameters, PDs are surveys of all geometrical features present in an AFM image, thus retaining all the information present in the topography measurement.

In Figure 3, the evolution with the reaction time of the PDs of the studied samples is presented. For a comparison between the morphology of the samples, the PDs of PDA/Au and NPs onto PDA/Au films are superimposed at each reaction time. PDs of PDA/Au samples showed that the PDA topmost layer appeared as formed by islands, mainly well separated from each other, and grew with increasing reaction times. A qualitative picture of the uptake of the magnetic nanoparticles is evidenced in the shape of the PDs in Figure 3. With a very short time (10 min), a limited number of very large (high) NP aggregates were observed in the 0-dimensional PD, while the PDA showed no relevant surface features. Further, it is possible to observe a cluster of scattered points, located in the proximity of the bisector. These could be attributed to irregularities and smaller clusters spread on the PDA/Au surface as well as around the bigger aggregates. From 20 min on, in the 0-dimensional PD, the NPs features started to cluster around the surface features formed by the PDA layer. In addition, we observed that at 20 min, some big NP aggregates, much taller than the PDA features, were still present, while, at 60 min and 24 h, the highest features in the 0-dimensional PD did not differ consistently between PDA/Au and NPs on PDA/Au samples. For 60 min and 24 h, the spreading of clusters of points of NPs around PDA islands increased. We interpret these results observing that NP uptake took place and was augmented along the surface features forming during PDA reactions, except at short times, where their absence left space for the uptake of larger aggregates, probably retained on the surface due to the different chemical composition at the early stage of the polymerization reaction. As a consequence of this picture, we found also in the 1-dimensional PD a change in the distribution of the NPs’ topological features, which assumed a round shape at 60 min and 24 h, thus indicating a larger degree of bundling with deeper valleys.

### 3.2. XPS Results

Since AFM could not provide direct information about the chemical composition of the features observed in the images, the uptake of Fe_3_O_4_ NPs on PDA/Au films was monitored by means of XPS. The survey spectra of the samples prepared by the deposition of Fe_3_O_4_ NPs on PDA/Au films prepared with various reaction times are shown in Figure 4. The areas of the Au4f, N1s, and Fe2p peaks as a function of the reaction time are shown in Figure 5. Upon increasing the reaction time, there was a decrease in the Au 4f peaks’ intensity due to the attenuation of the substrate peaks produced by the PDA film as well as by the Fe_3_O_4_ NPs adhered to it. From the attenuation of the Au4f peaks due to the PDA film, the average thickness of the polymer layer could be estimated. With an attenuation length of the Au 4f photoelectrons equal to 3 nm [30], the thickness of the PDA film after 24 h of reaction was found to be around 6 nm. This thickness was much lower than the average height of the islands in the AFM images. A possible explanation of this result is that a fraction of the substrate surface was uncovered or covered by a thin PDA layer, on top of which, islands of much larger height were present. Those were the features observed in the AFM images. The N1s peak intensity showed the expected trend, that is, an increase with reaction time. On the contrary, the Fe2p peak intensity did not vary monotonically vs. reaction time. This behavior was observed more clearly by plotting the Fe2p/N1s peak area ratios (Figure 5). A minimum uptake of Fe_3_O_4_ NPs was observed for the 20 min sample. By means of XPS, it was possible to determine the variation in the surface composition as a function of the reaction time in order to verify the explanation put forward to explain the AFM results, that is, the lack of NP adhesion on the surface covered by the intermediate species. For this purpose, we analyzed the C1s, N1s, O1s, and Fe2p peaks. The N1s spectra of the samples prepared with different reaction times and the results of their curve fitting analysis are shown in Figure 6.

In the curve fitting of the N1s spectra, the number of components was limited to three (two in the case of the 10 min sample), taking into account the energy resolution of the spectra and signal-to-noise ratio, particularly at the lowest PDA coverages. In the spectrum of the PDA film without Fe_3_O_4_ NPs, the main component at 400.3 eV could be ascribed to R-NH_2_ and R-NH-R groups (Figure 6, red line), the component at 399.6 eV to C=N-H groups (Figure 6, green line), and the one at 402.2 eV (Figure 6, blue line) to protonated R-NH_2_ groups [19,31,32]. Nitrogen atoms in the various intermediate species involved in the formation of PDA were expected to have similar Bes, resulting in overlapping components [33]. Under our experimental conditions, these components could not be separated by curve fitting analysis to obtain quantitative results. Nevertheless, the variations in the areas of the components as well as their positions indicated a change in the chemical composition of the PDA layer with reaction time. The change in surface chemical composition could have been responsible for the different uptake of Fe_3_O_4_ NPs on the surfaces prepared with different reaction times. Variations with reaction time were also observed for the O1s peak (Figure 7).

The O1s peak of the PDA could be fitted with two components: one peaked at 531.8 eV due to C=O groups (blue line) and the other at 533.2 eV, ascribable to C-O-H groups (red line) [19]. The intensity variation of these two components was consistent with the relative increase in quinone groups with respect to catechol groups in the course of the reaction. After the deposition of Fe_3_O_4_ NPs, an additional component with a maximum at ca. 530 eV due to the oxygen atoms in magnetite was present (cyan line) [34,35].

The position and shape of the Fe2p peaks did not change when magnetite NPs were deposited on PDA prepared with various reaction times (Appendix A). The position of the Fe2p_3/2_ peak (711.2 eV) was in the range of BEs reported for Fe_3_O_4_ NPs ([25] and references therein). A curve fitting analysis of the Fe2p peaks was rather awkward because of the overlapping with shake-up peaks originating from iron ions in the two formal oxidation states [36,37]. An attempted curve fitting analysis of the Fe2p spectrum of magnetite NPs on the PDA/Au film after 1 h of reaction time is shown in Appendix A. The curve fitting analysis of the C1s peaks revealed three main components (Appendix A). The component at a BE of 285.0 eV corresponded to carbon in hydrocarbons, that at 286.5 eV to C-O and C-N groups, and the third at ca. 288.5 eV to C=O [19,20,31,32,33]. The components due to C-O (C-N) and C=O showed relatively small changes in position and intensity upon increasing the reaction times (Appendix A).

## 4. Conclusions

By means of AFM and XPS techniques, we were able to monitor the uptake of magnetite NPs on PDA/Au as a function of the reaction time producing the PDA coating. AFM images showed that PDA grew on gold surfaces, forming islands with different heights and a roughness that increases with reaction time. The adhesion of Fe_3_O_4_ NPs on the gold surfaces coated with PDA was revealed by an increase in surface roughness and average height in the AFM images. The XPS results indicated that the amount of Fe_3_O_4_ NPs did not vary monotonically with the reaction time but reached a minimum in the early stages of the process. These results can be explained by considering the changes in the chemical composition and roughness of the film covering the gold surface during the reaction, which influenced its ability to bind Fe_3_O_4_ NPs. We demonstrated that by varying the reaction time to form the PDA coating, it is possible to control its ability to bind Fe_3_O_4_ NPs.

## Data Availability

The original contributions presented in the study are included in the article/Appendix A, further inquiries can be directed to the corresponding author/s.

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
