# Peer review of "Uptake of Magnetite Nanoparticles on Polydopamine Films Deposited on Gold Surfaces: A Study by AFM and XPS"

_nanomaterials, 2024, doi:10.3390/nano14211699_

Round 1
Reviewer 1 Report
Comments and Suggestions for Authors
General remarks
The manuscript (MS) submitted by Atrei et al. with the title of „Uptake of magnetite nnoparticles …” delivers practical information on the time-dependent adsorption probability of the magnetite nanoparticles on a polydopamine covered Au surfaces using AFM and XPS methods. The work contains also scientific analysis and explanation of the adsorption mechanism and its time dependent change. The work is well written. It is capable for publication in the present form, nevertheless, the authors should improve the manuscript by taken into account the remarks listed below.
a) Figure 1: the bottom image in the left column has a typical double-tip character. Have not you a better quality image? Moreover, the third image (from the top) in the right column is also not the best quality image.
b) The result obtained by the different methods (AFM, XPS) after the reaction time of „20 min” is not fitting harmoniously to the large time scale changes (it is noisier than others – Fig2S, or does not show characteristic large scale tendency – Fig 4, Fig 6, Fig 7), nevertheless, the authors explain this feature in details in the text (page 8-9). Actually, the 0-60 min period would be sufficient for plotting and it would have been better to measure two further points at 5 min and 30 min. Moreover, I suggest to plot (if possible) also the results obtained at zero min (0 min) reaction time (so, the initial point of the reaction), it would be also informative.
Special remarks
1. page 1 - check the affiliations in the case of JT and GDF;
2. page 1, row 16-19: in these sentences the „reaction time” relates tho the reaction time of the pretreatment performed during the formation of polydopamine layer. For clarity, please supplement these sentences.
3. Fig 1 and Fig 3: the visibility of the scales should be improved;
4. Fig 5 : in the top inset – Au 4f instead of Pt 4f should be written;
5. page 6, row 169: instead of „PH” --- „Persistent Homology (PH)” shoulsd be written;
6. Fig 2, Fig 5, Fig 6: x-axis label – Reaction time (min)” would be better and inside the figures insttead of „min.” should be „min”;
7. Fig 1: in the figure legend it should be „(a) 10 minutes; (b) 20 minutes; (c) 60 minutes; (d) 24 hours”;
8. page 11, row 270: instead of „producing the coating”, it would be more clear „producing PDA-coating”;
9. page 11 – author contributions, JT exciplicit contrubution is missing, although, he belongs to „all authors”;
Author Response
The manuscript was revised according to the comments of the reviewers (see below a list of the comments and our replies). Changes of the text are marked in yellow. During the revision we found that the section on the persistence homology analysis was not very clear. Hence we expanded this part and modified Figure 3 trying to better describe the method and the results.
Reviewer #1.
a) Figure 1: the bottom image in the left column has a typical double-tip character. Have not you a better quality image? Moreover, the third image (from the top) in the right column is also not the best quality image.We agree with the referee about the quality of the AFM images that in some cases is not the state of the art. The best quality images were selected. Although some images show double tip effects, this does not influence the conclusions derived from the AFM images. The bottom image in the left column has been changed with another one measured for a sample prepared under the same conditions but with a new tip.
b) The result obtained by the different methods (AFM, XPS) after the reaction time of „20 min” is not fitting harmoniously to the large time scale changes (it is noisier than others – Fig2S, or does not show characteristic large scale tendency – Fig 4, Fig 6, Fig 7), nevertheless, the authors explain this feature in details in the text (page 8-9). Actually, the 0-60 min period would be sufficient for plotting and it would have been better to measure two further points at 5 min and 30 min. Moreover, I suggest to plot (if possible) also the results obtained at zero min (0 min) reaction time (so, the initial point of the reaction), it would be also informative.
The average height, surface roughness derived by analysing the AFM images of Fe3O4 NPs deposited on PDA/Au films do not vary monotonously with the reaction time of PDA film formation. The amount of Fe3O4 NPs adhered to the surface is minimum with a reaction time of 20 minutes. This may be the reason why it seems that the data at 10 and 20 minutes do not fit harmoniously to the large time scale changes. Although with more points measured at intermediate reaction times a more detailed picture would have be obtained, the reported data provide a trend of the effect of the reaction time on surface morphology, composition and capacity to bind Fe3O4 NPs of the PDA film. The parameters derived by the analysis of the AFM images of the clean Au substrate (i.e. reaction time equal to zero) are reported in the text (page 5, lines 1 and 2). These points would be hardly visible in the plot of the data in Figure 2.
Special remarks
1.page 1 - check the affiliations in the case of JT and GDF;
Checked the affiliations and inserted more details in that of GDF.
2.page 1, row 16-19: in these sentences the „reaction time” relates tho the reaction time of the pretreatment performed during the formation of polydopamine layer. For clarity, please supplement these sentences.
We modified the abstract
3.Fig 1 and Fig 3: the visibility of the scales should be improved;
Figure 1 contains several images and it would be hard to increase the size of the scales without reducing that of the images.
4.Fig 5 : in the top inset – Au 4f instead of Pt 4f should be written;
The mistake was corrected.
5.page 6, row 169: instead of „PH” --- „Persistent Homology (PH)” shoulsd be written;
Changed the text.
6.Fig 2, Fig 5, Fig 6: x-axis label – Reaction time (min)” would be better and inside the figures insttead of „min.” should be „min”;
Changed the figures.
7.Fig 1: in the figure legend it should be „(a) 10 minutes; (b) 20 minutes; (c) 60 minutes; (d) 24 hours”;
Changed the Figure caption.
8.page 11, row 270: instead of „producing the coating”, it would be more clear „producing PDA-coating”;
Changed the text.
9.page 11 – author contributions, JT exciplicit contrubution is missing, although, he belongs to „all authors”;
JT performed the XPS measurements. His contribution is included in “Investigation”.
Reviewer 2 Report
Comments and Suggestions for Authors
The authors investigated the adsorption behavior of magnetite nanoparticles on polydopamine films on gold surfaces and characterized them using AFM and XPS techniques. The results showed that the preparation time and chemical composition of the PDA films had a significant effect on the adsorption. However this work has some weaknesses that need to be improved. The following is what I think needs to be revised accordingly:
1. The abstract section needs to be revised.It is recommended that the purpose of the study be stated more clearly in the abstract, for example to explore the effect of preparation time of PDA films on their ability to bind to Fe3O4 nanoparticles.
2. In the introduction section other types of NPs are mentioned, is it possible to include a generalized comparison of the other types of NPs and a description of the unique advantages of magnetic NPs.
3. It is suggested to add the current research status of PDA films in the field of nanoparticle adsorption and briefly introduce the application of AFM and XPS techniques in the characterization of nanomaterials.
4. It is recommended to provide more detailed conditions for the preparation of PDA films and characterization data of magnetite nanoparticles, such as particle size distribution, surface morphology, and chemical composition.
5. Additional analysis of the adsorption mechanism of Fe3O4 nanoparticles on PDA films is recommended, e.g., considering the relative contributions of chemisorption and physisorption.
6. It is recommended to keep the valid digits after the decimal point consistent in the article. For example, in page 3, line 87,“pH 8.0”not“pH 8”.
7. Some details in the article and some English expressions need to be considered to achieve better expression effect. For example, in page 3, line 78, An extra comma is added; in page 3, line 94, There's an extra“ca”over there.
8. The conclusion part of the article contains some vague and uncritical expressions, and does not clearly state the specific method of controlling the adsorption amount, as well as the change rule of adsorption amount corresponding to different reaction times.

Author Response
The manuscript was revised according to the comments of the reviewers. Changes of the text are marked in yellow. During the revision we found that the section on the persistence homology analysis was not very clear. Hence we expanded this part and modified Figure 3 trying to better describe the method and the results.
The authors investigated the adsorption behavior of magnetite nanoparticles on polydopamine films on gold surfaces and characterized them using AFM and XPS techniques. The results showed that the preparation time and chemical composition of the PDA films had a significant effect on the adsorption. However this work has some weaknesses that need to be improved. The following is what I think needs to be revised accordingly:
- The abstract section needs to be revised.It is recommended that the purpose of the study be stated more clearly in the abstract, for example to explore the effect of preparation time of PDA films on their ability to bind to Fe3O4 nanoparticles.
As suggested by the reviewer the abstract has been modified to state more clearly the aim of the study. - In the introduction section other types of NPs are mentioned, is it possible to include a generalized comparison of the other types of NPs and a description of the unique advantages of magnetic NPs.
In the literature there are numerous studies on nanoparticles of a variety of materials coated with PDA. Among these a large fraction concerns magnetic nanoparticles. A review of these studies would be beyond the aim of this paper. However, the readers interested can deep the subject referring to the cited references are cited (see refs. 14-18].
3. It is suggested to add the current research status of PDA films in the field of nanoparticle adsorption and briefly introduce the application of AFM and XPS techniques in the characterization of nanomaterials
As discussed above, such survey on the research status of PDA films would be out of the aim of the present work. The AFM and XPS are well established techniques in the field of nanomaterials. Hence, it seems there is no need to introduce the methods (besides the persistent homology method which is a new approach to analyse AFM images).
4. It is recommended to provide more detailed conditions for the preparation of PDA films and characterization data of magnetite nanoparticles, such as particle size distribution, surface morphology, and chemical composition
The characterization of the magnetite nanoparticles is reported in detail in refs. 24 and 25. A sentence has been added about this point.
5. Additional analysis of the adsorption mechanism of Fe3O4 nanoparticles on PDA films is recommended, e.g., considering the relative contributions of chemisorption and physisorption.
On the basis of the results of this work is not possible to derived any conclusion about the nature of the interaction between PDA and the NPs. Although the details about the structure of PDA are unknown, according to the most probable models PDA films present various functional groups capable to bind to magnetite nanoparticles. Hence, it may be that magnetite nanoparticles are chemisorbed.
6. It is recommended to keep the valid digits after the decimal point consistent in the article. For example, in page 3, line 87,“pH 8.0”not“pH 8”.
The reported numbers have been corrected.
7. Some details in the article and some English expressions need to be considered to achieve better expression effect. For example, in page 3, line 78, An extra comma is added; in page 3, line 94, There's an extra“ca”over there.
The text has been thoroughly revised. These and other errors have been corrected.
Reviewer 3 Report
Comments and Suggestions for Authors
The authors have reported a study, devoted to magnetite NPs deposition onto gold surface, via polydopamine coating approach.
There have been used AFM and powder XRD methods to determine morphology of the deposited NPs. The oxidation reactions of the coating agent have been examined; thus determining its binding blocks.
The molecular levels mechanistic aspects of the processes have been detailed on studying the binding energy data. The reactions kinetics has been determined, as well.
There are minor corrections needed in order to motivate better the study, because of currently there is only mentioned that the functionalization of the gold surface has potential practical applications to the environmental and biomedical research fields (row 33.)
However, there should be briefly detailed on, how such paterials would find applications to the discussed interdisciplinary fields, and why namely polydopamine is coating agent of choice for these potential purposes of applications.
In addition, there should be corrected in Figure 1: Instead of DHI (tautomer), there should be 'LDAC (tautomer)'.
The paper also suffers from grammatical English errors, particularly highlighting prepositional phrasal verbs. The authors could spend time in the latter task, as well.
Moderate English editing is needed.
Author Response
The manuscript was revised according to the comments of the reviewers. Changes of the text are marked in yellow. During the revision we found that the section on the persistence homology analysis was not very clear. Hence we expanded this part and modified Figure 3 trying to better describe the method and the results.
There are minor corrections needed in order to motivate better the study, because of currently there is only mentioned that the functionalization of the gold surface has potential practical applications to the environmental and biomedical research fields (row 33.)
However, there should be briefly detailed on, how such paterials would find applications to the discussed interdisciplinary fields, and why namely polydopamine is coating agent of choice for these potential purposes of applications.
The most relevant feature of PDA is its capacity to adhere to surfaces of many different materials. The functional groups (cathecol, quinone, amine) present in PDA films allow the functionalization with a variety of molecules. Moreover, thanks to its adhesive properties, PDA films can be used as a glue to anchor nanoparticles on solid surfaces. As far as the environmental applications are concerned, nanoparticles (in particular magnetic ones) coated with PDA (opportunely functionalized) can adsorb organic molecules, heavy metals from polluted waters. To avoid to lengthen the introduction we refer the readers to the cited papers.
In addition, there should be corrected in Figure 1:Instead of DHI (tautomer), there should be 'LDAC (tautomer)'.
The molecule reported in Figure 1 (the fifth) is the tautomer of DHI (the fourth molecule) not of LDAC (the third molecule).
The paper also suffers from grammatical English errors, particularly highlighting prepositional phrasal verbs. The authors could spend time in the latter task, as well.
We thoroughly checked the text for typos and grammatical English errors.
Round 2
Reviewer 1 Report
Comments and Suggestions for Authors
I accept readily the response of the authors.